# Latest Advances in Endoscopic Detection of Oesophageal and Gastric Neoplasia

**DOI:** 10.3390/diagnostics14030301

**Published:** 2024-01-30

**Authors:** William Waddingham, David G. Graham, Matthew R. Banks

**Affiliations:** 1Department of Gastroenterology, Royal Free London NHS Foundation Trust, London NW3 2QG, UK; 2Department of Gastroenterology, University College London NHS Foundation Trust, London NW1 2BU, UK

**Keywords:** Barrett’s oesophagus, squamous dysplasia, chronic atrophic gastritis, intestinal metaplasia, surveillance

## Abstract

Endoscopy is the gold standard for the diagnosis of cancers and cancer precursors in the oesophagus and stomach. Early detection of upper GI cancers requires high-quality endoscopy and awareness of the subtle features these lesions carry. Endoscopists performing surveillance of high-risk patients including those with Barrett’s oesophagus, previous squamous neoplasia or chronic atrophic gastritis should be familiar with endoscopic features, classification systems and sampling techniques to maximise the detection of early cancer. In this article, we review the current approach to diagnosis of these conditions and the latest advanced imaging and diagnostic techniques.

## 1. Introduction

Oesophageal and gastric cancers remain major worldwide causes of cancer-related deaths, with oesophageal cancer ranking seventh in incidence and sixth for mortality, and gastric cancer ranking fifth for incidence and fourth for mortality globally [1]. There is considerable geographical variation in the occurrence of these cancers. Concerningly, across developed countries, the incidence of oesophageal adenocarcinoma (OAC) is rising and is predicted to continue to do so until at least 2040 [2]. This has been suggested to be related to increasing levels of obesity and gastro-oesophageal reflux, and perhaps reduced prevalence of *H. pylori* [1]. Notably, despite global declines in *H. pylori* infection, recent studies have also shown an increase in the incidence of stomach cancer (cardia and non-cardia) among adults aged <50 years in both low-risk countries (including the United States, Canada and the United Kingdom) and high-risk countries. It has been hypothesised that this increase may be accounted for by a rising prevalence of autoimmune chronic gastritis and dysbiosis of the gastric microbiome [1].

Late diagnosis of oesophageal and gastric cancers continues to contribute to the poor outcomes associated with these diseases, with UK ten-year survival figures as low as 12.4% for oesophageal and 16.7% for gastric cancer [3]. Further to this, the COVID-19 pandemic has led to significant delays in diagnosis, with one study estimating these delays have caused a 6% increase in oesophageal cancer deaths [4]. Early diagnosis and treatment is a key determinant in improving outcomes; early gastric cancer (EGC), for example, has a good prognosis, with a five-year survival rate of between 69% and 82% [5]. Furthermore, when these cancers are confined to the mucosa (stage T1a) endoscopic resection (ER) is often curative and associated with excellent outcomes, with 5-year survival has high as 92% [6].

Endoscopy is the accepted gold standard test for diagnosis of oesophageal and gastric cancers, allowing direct visualisation and sampling for histological confirmation. Upper endoscopy includes both standard transoral endoscopy (oesophago-gastroduodenoscopy, OGD) and trans-nasal endoscopy (TNE), the latter of which can be performed awake in the outpatient clinic setting. Both of these diagnostic techniques are safe, well tolerated and provide a high degree of diagnostic accuracy in the hands of an appropriately trained endoscopist. Both oesophageal and gastric adenocarcinoma have endoscopically detectable precursors, which carry an increased risk of developing malignancy, and it is on this basis that guidelines for endoscopic surveillance of higher-risk patients have been developed, with the aim of detecting cancer earlier. The purpose of this review is to summarise the latest evidence on endoscopic detection including advanced imaging modalities and surveillance of oesophageal and gastric neoplasia.

## 2. High-Quality Upper GI Endoscopy

The way we perform upper GI endoscopy is fundamental to our ability to diagnose neoplasia early and avoid missed diagnoses. Missed upper GI cancers, or post-endoscopy upper GI cancers (PEUGIC) are defined as a cancer diagnosis occurring within 3 years of a previously negative endoscopy. A 2014 meta-analysis found overall mis-rates of 11.3% for upper GI cancers at 3 years [7]. More recently a Japanese study found that detection rates of EGC vary considerably by endoscopist (0.09–2.87%), and endoscopists with higher EGC detection rates were better at detecting minute EGCs (<5 mm) [8]. Awareness of the diagnostic limitations of upper GI endoscopy is essential and requires attention through training, audit and honest appraisal of practice. The British Society of Gastroenterology (BSG) has published a position paper outlining key performance indicators (KPIs) for upper GI endoscopy [9]. Specifically, this includes ensuring sufficient time is allocated to procedures, with adequate mucosal visualisation, through a combination of air insufflation, aspiration and the use of mucosal cleansing agents (e.g., simethicone or n-acetylcysteine). An OGD should include assessment and photo-documentation of all anatomical landmarks, with a total of eight photographs recommended. It is also advised that the inspection time for surveillance procedures, e.g., Barrett’s oesophagus and gastric atrophy or metaplasia, should be recorded, and a minimum examination time of 7 min is advised [10]. Units should have mechanisms in place for managing timely surveillance of high-risk patients, and for auditing rates of PEUGIC.

## 3. Multi-Disciplinary Approach

Management of upper GI cancers requires a multidisciplinary team (MDT) approach, including gastroenterologists, radiologists, surgeons, pathologists, oncologists and specialist nurses. A similar model is equally applicable to the management of Barrett’s neoplasia, squamous neoplasia and gastric neoplasia. MDT meetings authorise decisions about endoscopic resection vs. surgery, and ensure appropriate audit and follow-up of cases occurs. Patients with Barrett’s segments of ≥10 cm, a confirmed diagnosis of low-grade dysplasia, high-grade dysplasia or early cancer should all be referred to a specialist centre and management decisions ratified through an MDT [11,12].

## 4. Barrett’s Oesophagus

### 4.1. Definition

Barrett’s oesophagus (BE) is an acquired, premalignant condition occurring in response to gastro-oesophageal reflux and leading to replacement of the normal squamous mucosa with a columnar lined distal oesophagus [13]. There have been several differing definitions histologically and endoscopically over the years since its identification. The condition is defined by the British Society of Gastroenterology (BSG) guidelines as “an oesophagus in which any portion of the normal distal squamous epithelial lining has been replaced by metaplastic columnar epithelium, which is clearly visible endoscopically (≥1 cm) above the GOJ and confirmed histopathologically from oesophageal biopsies” [13]. This previously differed from the American guidelines (ASGE and AGA), which defined Barrett’s as the presence of intestinal metaplasia of the tubular oesophagus, making no cut-off of length requirement for diagnosis [14], although a more recent 2022 updated American guideline now acknowledges that at least 1 cm of length of columnar mucosa is required for a BE diagnosis [15].

### 4.2. Barrett’s Oesophagus: Diagnosis—Endoscopy

The gold standard diagnostic tool for Barrett’s oesophagus is endoscopy; either OGD or TNE are equally sensitive and specific for diagnosis [16]. However, formal diagnosis requires a combination of endoscopic features and histological confirmation. In order to accurately identify a Barrett’s segment at endoscopy, it is essential to first locate the gastro-oesophageal junction (GOJ); this should be done by locating the proximal end of the gastric folds, done with minimal air insufflation [13]. This allows distinction between an irregular Z-line, i.e., tongues of columnar lined mucosa <1 cm in length and no circumferential columnar mucosa, vs. Barrett’s, which, by definition, is ≥1 cm in length. The recent Kyoto consensus on the anatomy, pathophysiology and clinical significance of the gastro- oesophageal junction recommended that the junction be redefined as that 1 cm above and below the distal end of the palisade vessels [17]. Any columnar epithelium above the middle of this zone is considered to be Barrett’s epithelium. At white light endoscopy, columnar lined Barrett’s is redder in colour, or salmon-pink, and coarser in appearance, with a colour more in keeping with the stomach, whereas squamous mucosa appears paler and smoother.

Once a Barrett’s segment has been identified endoscopically, it should be measured using the Prague classification (Figure 1), where C = the length of circumferential Barrett’s from the GOJ and M = the maximal length, i.e., the most proximal point of any of the tongues [18]. As well as describing the extent of BE and any islands proximally, the location of the squamo-columnar junction, diaphragmatic pinch and any hiatus hernia present should also be documented. Despite the validated Prague classification system, overdiagnosis remains a prevalent issue; in one retrospective study, 32.3% of patients with a previous diagnosis of BE had their diagnosis revised to no BE after a consensus review [19]. As such, biopsies at endoscopy should not be routinely taken for irregular Z-lines or non-circumferential tongues of CLO <1 cm [13,20].

Biopsies for suspected BE should be taken using the Seattle protocol (Figure 1). This involves four quadrantic biopsies starting 1–2 cm above the GOJ and taken at every 2 cm, with distal biopsies taken first, then advancing proximally to minimise obscuring of views by bleeding [13]. Additionally, if a visible lesion is present or there are areas of concern with the segment, targeted biopsies should be taken first and sent in a separate pot. For visible lesions, the Paris classification should be used to standardise documentation of size and morphology.

### 4.3. Barrett’s Oesophagus: Surveillance and Cancer Risk

The aim of endoscopic surveillance is to detect cancer at a stage when intervention can be curative. In the setting of BE, surveillance should detect neoplasia before submucosal invasion, when the risk of lymph node metastases increases significantly (9–50%) [21,22,23,24]. The annual cancer conversion rate for non-dysplastic BE has been estimated at 0.22–0.5%/year. Although there are conflicting data, the presence histologically of IM has been shown to confer an additional increased risk of cancer; in a Northern Ireland registry study, the annual incidence of HGD and cancer in patients with IM was significantly higher than in those without IM (0.38% vs. 0.07%) [25]. There is a lack of RCT evidence for survival or mortality benefits with BE surveillance; however, guidelines are based on evidence that surveillance correlates with an earlier stage of cancer diagnosis and improved cancer survival. It is, therefore, recommended [13].

For non-dysplastic BE, surveillance should take into account the presence of IM and the length of the BE segment, as well as patient fitness. Surveillance is not recommended for IM at the cardia, or for BE segments <3 cm with no IM on Seattle biopsies on two consecutive endoscopies [13]. For patients with BE of less than 3 cm with IM, they should be offered endoscopic surveillance every 3–5 years. For BE of 3 cm or longer, patients should receive surveillance endoscopy every 2–3 years, and consideration of referral to a specialist centre is advised for very long segments >10 cm [13]. During surveillance endoscopy, the endoscopist should aim to spend a minimum of 1 min inspection time per 1 cm of circumferential BE [10].

It is known that a diagnosis of Barrett’s oesophagus carries a reduced health-related quality of life, and this likely includes worries about cancer risk, endoscopic surveillance, and symptom control. Endoscopic therapy for Barrett’s dysplasia is not associated with improved quality-of-life measures, including when considering similar levels of cancer worry pre- and post-treatment [26]. There is a need to improve the health-related quality of life in Barrett’s oesophagus. Communication with patients in clinic is necessary to understand the psychological and clinical burden this disease carries and to try to reassure patients and alleviate this.

### 4.4. Endoscopic Detection of Barrett’s Dysplasia: Chromoendoscopy

All endoscopic surveillance should utilise high-definition endoscopy. Chromoendoscopy dye sprays such as methylene blue and indigo carmine have been investigated extensively as diagnostic adjuncts, but a meta-analysis showed no incremental diagnostic yield of methylene blue-targeted biopsies for dysplasia diagnosis in BE [27]. Acetic acid (AA) chromoendoscopy (at 1.5–2.5% concentration) stains BE white (Figure 2A) due to breakage of glycoprotein disulphide bonds in the superficial mucus layer and acetylation of cellular proteins [28]. Dysplasia loses this whitening effect earlier than non-dysplastic BE, which is referred to as early loss of aceto-whitening. Additionally, AA helps improve the definition of the mucosal architecture, thus helping to display irregular or distorted pits that are indicators of the presence of dysplasia. Two recent meta-analyses showed very high levels of accuracy in the detection of dysplasia (HGD) and intramucosal cancer (IMCa) with AA: pooled sensitivity, negative predictive value and specificity for AA chromoendoscopy were 96.6% (95% confidence interval (CI), 95–98), 98.3% (95% CI, 94.8–99.4) and 84.6% (95% CI, 68.5–93.2), respectively [29,30]. AA is therefore a useful and widely available adjunct that should be used in BE surveillance to aid dysplasia detection.

### 4.5. Endoscopic Detection of Barrett’s Dysplasia: Advanced Imaging

A number of advanced imaging modalities are now widely available (narrow-band imaging, NBI—Olympus, iScan—Pentax, Fujinon intelligent colour enhancement (FICE)—Fujinon), and use optical light filters, often with downstream processing to enhance visualisation of the mucosal surface (Figure 2B,C). Many centres will work from a single endoscope platform and, therefore, would tend to only use the advanced imaging modality that is available in the department. Advanced imaging technologies have been evaluated in the context of improving dysplasia detection in BE, with most data relating to NBI. At the time of European and British guidelines being published, the routine use of electronic chromoendoscopy was not advised, due to the lack of evidence of its additional value over high-definition white light endoscopy (WLE) [11,13]. However, an updated meta-analysis including 504 patients found virtual chromoendoscopy with HD-WLE was associated with a higher detection rate of HGD/OAC compared with HD-WLE alone (14.7% vs. 10.1%; relative risk, 1.44) [31]. A further meta-analysis found that NBI with targeted biopsies had a >90% sensitivity and specificity for detection of BE neoplasia [29]. As such, more recent American guidelines (the ASGE and AGA technology committee) now recommend the routine use of HD-WLE and virtual chromoendoscopy [14,32]. The Barrett’s International NBI Group (BING) developed and validated a classification system for identifying HGD and early adenocarcinoma using NBI; this includes both mucosal and vascular patterns under NBI, and classifies them into regular or irregular, resulting in high accuracy (85%) and inter-observer agreement (kappa 0.681) [33]. There is, therefore, a strong body of evidence to advocate for the routine use of NBI as an adjunct to Seattle protocol biopsies for all BE surveillance cases.

### 4.6. Endoscopic Detection of Barrett’s Dysplasia: Confocal Laser Endomicroscopy (CLE) and Volumetric Laser Endomicroscopy (VLE)

CLE uses blue laser light combined with an intravenous fluorescent agent to provide in vivo mucosal inspection at microscopic levels (up to 1000× magnification). There are two types of CLE—endoscope-based (eCLE) and probe-based (pCLE). eCLE was shown in a multi-centre international RCT to significantly increase the diagnostic yield of HGD and early adenocarcinoma vs. Seattle biopsies and to improve real-time decision making for endoscopic therapy [34]. The ASGE technology committee’s updated systematic review and meta-analysis found a pooled sensitivity, NPV and specificity of 90.4%, 98.3% and 92.7% (95% CI, 87–96), respectively, meeting the PIVI thresholds [29]. This meta-analysis also analysed the performance of pCLE, revealing a sensitivity of 90.3%, NPV of 95.1% and specificity of 77%, falling just below PIVI thresholds. At present, CLE is not widely available even in specialist centres and, thus, is yet to be adopted further, nor is it recommended for routine use by guidelines. It also requires the use of an IV fluorescent agent and a longer procedure time.

VLE is a form of optical coherence tomography (OCT) that provides a complete scan of the oesophageal wall. It utilises infra-red light to image the surface and deeper layers of tissue, producing a cross-sectional, high-resolution (up to 7µm), real-time image. It is reported to be able to scan a 6 cm length of oesophagus in 90 s [35]. VLE has been shown to improve dysplasia detection over Seattle biopsies in a post-dysplasia treatment cohort (8.3% vs. 32.7%, *p* = 0.02), whereas in a treatment-naïve BE cohort, there was no difference. More recently, a study has looked at the feasibility of VLE in combination with computer-aided detection (CAD) of BE dysplasia, and found higher levels of BE neoplasia detection in a retrospective analysis [36]. At the time of the ASGE technology appraisal, as there was only one study of VLE in BE, there was insufficient evidence for recommending its use in BE surveillance [29]. While it is important to be aware of these technologies, they are not available in most centres, and remain in use primarily as part of research studies.

## 5. Alternative Sampling Technologies

### 5.1. Wide-Area Transepithelial Sampling (WATS)

WATS is an alternative epithelial sampling technique aimed at reducing the limitations of the gold standard, forceps biopsy. Traditional Seattle protocol forceps biopsies are time-consuming, operator-dependent and poorly adhered to. It has also been shown that quadrantic biopsies of a BE segment sample as little as 3.5% of the surface of the segment [37]. Although this can be mitigated to some extent by the use of enhanced imaging endoscopy, there remains a significant operator dependency when taking representative samples with standard biopsies. WATS utilises a brush-biopsy technique, wherein an abrasive through the scope brush samples the entire thickness of the epithelial layer, allowing efficient sampling of a wide area of the oesophagus. Analysis of the resulting tissue specimen is aided by a neural network-based, computer-assisted scan of each slide with a three-dimensional reconstruction (WATS-3D) that is displayed to the pathologist on a video monitor.

Data to support the use of WATS have led to it being incorporated into the ASGE and AGA Barrett’s surveillance guidelines as an adjunct to Seattle biopsies for dysplasia detection [14,32]. A recent systematic review and meta-analysis of seven studies showed an increased yield for dysplasia detection of 7.2% of WATS over forceps biopsies [38]. In addition, the histopathological interpretation of WATS specimens was shown to have less interobserver variability than standard histopathology, with a kappa of 0.86 [39], making the use of WATS a reliable diagnostic tool.

### 5.2. Cytosponge

A number of studies have investigated the use of a swallowed sponge device to detect BE. The Cytosponge is a small mesh sponge attached to a string, within a soluble capsule. It can be safely swallowed orally, under the instruction of a nurse, in primary care settings to collect oesophageal epithelial samples for analysis. The device can collect cells from the entire oesophagus and has been reported as an accurate and acceptable method of sample collection for BE screening [40,41]. A UK multicentre RCT compared Cytosponge-TFF3 testing with the usual care in patients taking acid suppressants in primary care. The estimated adjusted relative risk of detecting BE was 10.6 (95% confidence interval, 6.0–18.8) for the Cytosponge-TFF3 group vs. the usual care in a 12-month follow-up, equating to 127 new BE cases (Cytosponge-TFF3) vs. 13 (usual care) [42]. The Cytosponge has yet to be validated for the surveillance of BE patients; however, several pilot studies have been carried out showing that the Cytosponge-TFF3 could enable targeted endoscopy for higher-risk individuals [43,44], with TFF3 positivity increasing with segment length (odds ratio 1.37 per 1 cm segment length), while those BE patients with TFF3-negative sponge results and short segments could potentially be saved endoscopic surveillance. Further longitudinal studies are needed to establish the Cytosponge’s role in BE surveillance and whether it can be used in a screening role outside of secondary care.

## 6. Oesophageal Squamous Neoplasia

### 6.1. Introduction and Diagnosis

Squamous cell carcinoma (SCC) remains the leading cause of oesophageal cancer outside of Western countries. Heavy drinking and smoking are risk factors in developed countries for SCC, with a RR of 2.62 for alcohol intake of 12.5 g/day or more and a RR of 2.63 for smoking. Additional suspected risk factors include betel nut chewing, pickled vegetables and very hot food or beverage consumption [1]. An awareness of risk factors for SCC is essential to better detect cases at endoscopy. Additionally, a history of head and neck SCCs was found to carry a risk of 3.2–9.9% for oesophageal SCC in screening studies [45,46]; there is, therefore, a strong argument for offering endoscopic screening to patients with head and neck SCC. As with adenocarcinoma, when patients present with symptoms of dysphagia, SCC is often diagnosed at an advanced stage, with poor survival rates. Superficial early-stage SCC is often asymptomatic and carries a much better survival rate if identified and treated.

### 6.2. Endoscopic Diagnosis of SCC

Early (intramucosal) SCC is often flat with minimal change to the shape of the oesophageal surface; this makes it difficult to detect with standard WLE. Meanwhile, more advanced SCC tends to be elevated (Paris I) or excavated/depressed (Paris III) or a combination of the two [47]. WLE alone is insufficient for reliably detecting early SCC, and the use of adjuncts or advanced imaging endoscopy is needed to maximise diagnostic yield in higher-risk patients.

### 6.3. Endoscopic Detection: Chromoendoscopy & Advanced Imaging

Lugol’s iodine chromoendoscopy (LCE) is an iodine-based spray that is deployed through the scope via a spray catheter. Iodine is taken up by glycogen in normal squamous mucosa, whereas areas of SCC are glycogen-poor, resulting in Lugol voiding areas (Figure 2D) due to reduced uptake of iodine. This is also known as the “pink-colour sign” and correlates histologically with at least HGD [47]. Lugol-targeted biopsies in a high-risk cohort identified patients with squamous HGD with 46% sensitivity and 90% specificity [48]. A separate large cohort study found that >60% of Lugol voiding lesions >10 mm in size had at least HGD, whereas <5% of voiding lesions <5 mm contained HGD [49]. A more recent meta-analysis (12 studies, 1911 patients) looked at NBI and Lugol’s and found a sensitivity of 92% (95% CI, 86–96%) and specificity of 82% (95% CI, 80–85%) for Lugol’s chromoendoscopy in the per-patient analysis [50]. This same meta-analysis found NBI to be accurate for assessing squamous neoplasia (HGD and early SCC), with a sensitivity of 88%, specificity of 88% upon per-patient analysis, which, in fact, showed superiority over Lugol’s in terms of specificity.

The vessel pattern of squamous mucosa has been described as intrapapillary capillary loops (IPCLs) (Figure 2E,F). The Japanese Esophageal Society (JES) IPCL classification describes five patterns, correlating to normal (type I), inflamed (type II), inflamed/low-grade neoplasia (type III), low-grade neoplasia/high-grade neoplasia (type IV) and cancer (type V) [51]. NBI with magnification has been shown to accurately predict the depth of SCC invasion in expert hands (type V1 = m1, type V2 = m2, type V3 = m3/SM1, type Vn = SM2) [52]. Similar degrees of accuracy have been seen with FICE and magnified FICE in detecting early SCC compared to Lugol’s, 92.6% vs. 88.9% (*p* = 0.642), while the addition of magnification endoscopy did not significantly improve detection rates [53]. The JES has since developed a simplified classification system for use with magnification to estimate the invasion depth of superficial oesophageal SCC; this can be split into type A (non-cancerous) with normal IPCLs or abnormal microvessels without severe irregularities; and type B (cancerous) with abnormal microvessels with severe irregularities (Figure 2E,F). Type B is further subdivided into B1, B2 and B3, each corresponding to progressive depths of invasion (from T1a to T1b SM2) [54].

## 7. Gastric Atrophy and Intestinal Metaplasia

### 7.1. Definition

The development of gastric adenocarcinoma (non-cardia type) occurs in the context of Helicobacter pylori-related chronic inflammation, and less commonly in the context of autoimmune gastritis. The Correa cascade describes a stepwise transformation from chronic inflammation to chronic atrophic gastritis (CAG), intestinal metaplasia and finally dysplasia and cancer [55]. As such, CAG is considered a precursor lesion to gastric adenocarcinoma. Atrophic gastritis is defined histologically by the presence of chronic inflammation and the loss of pre-existent gastric glands, with native gastric glands replaced by metaplastic glands when IM is present [56].

### 7.2. Endoscopic Diagnosis of CAG and Gastric IM

At endoscopy, atrophy is readily identified with white light, by loss of the gastric rugae, mucosal pallor and increased visibility of mucosal vessels. The atrophic border can usually be identified as a demarcation between normal mucosa and atrophic mucosa; this moves proximally as the disease progresses. Using WLE, gastric IM appears as paler-white patches, or elevated plaques, surrounded by pinker areas of mucosa (Figure 3A). There is often an irregular uneven contour to the gastric surface, with patchy erythema a sign associated with gastric IM [57]. WLE should not be used as the sole modality for the detection and assessment of gastric IM due to its inferior accuracy compared to NBI (53% vs. 87%: *p* < 0.001) [58].

### 7.3. Gastric Intestinal Metaplasia: Advanced Imaging

A number of studies have shown improved detection of gastric IM with enhanced imaging (NBI). As patches of gastric IM expand, the glands elongate to form a “groove type pattern” similar to that of the antrum or villiform pattern of the intestine (Figure 3A,C). Although these changes can easily be distinguished from the normal corpus, IM in the antrum is more difficult to characterise [57]. Additional features of gastric IM that can aid the endoscopic diagnosis in the antrum include the light blue crest (LBC) and the marginal turbid band best seen with NBI and magnification [59,60]. The Endoscopic Grading of Gastric Intestinal Metaplasia (EGGIM) score is an alternative method of staging the stomach by the presence of gastric IM, and utilises NBI to assess each of the five areas of the stomach (<30% or >30% of the mucosal surface), with targeted biopsies to confirm. A recent validation study compared the EGGIM score with the histological OLGIM score and suggested that such an endoscopic staging system may be clinically efficacious [61].

### 7.4. Gastric Neoplasia Detection: Advanced Imaging

High-definition endoscopy should always be used when assessing the high-risk stomach. Features suggestive of dysplasia include irregular vessels and glands, with loss of the normal pits and mucosal pattern. Non-healing gastric ulcers are also a feature of neoplasia. The use of NBI with magnification (in conjunction with WLE) yields a higher accuracy for detection of early gastric cancer (EGC) with a sensitivity of 95% and specificity of 96.8% for depressed EGCs [62]. The Japanese Vascular Surface (VS) classification was proposed in 2009 [63] and describes the microvascular (MV) pattern and microsurface (MS) pattern independently to help predict the histology of EGC. Using the VS classification for EGC, features include the presence of a demarcation line and identification of irregular MV and MS patterns inside the demarcation line. The MV pattern can be further subdivided into two types: fine network pattern (with a mesh formation) and corkscrew pattern (tortuous pattern with no connections) [64]. Studies of the diagnostic performance of NBI with magnification using the VS classification system have shown up to 95% sensitivity and 96% specificity [65].

Endocytoscopy (EC) offers ultra-high magnification, allowing mucosal visualisation at the cellular level. EC has been studied in several studies to augment the detection of early, superficial gastric lesions. This includes the detection of signet ring carcinomas in the stomach and gastric lymphomas. There is as yet no standardised classification for EC and gastric lesion diagnosis [65]; additionally, EC is rarely available outside of research settings.

### 7.5. Gastric Surveillance

The grade and severity of CAG are predictive of GC risk, with population studies suggesting annual incidences of GC of 0.1–0.25% for CAG and gastric IM [66]. Longitudinal studies suggest that endoscopic staging of CAG with the modified Kimura–Takemoto classification system is a useful stratification system to predict GC risk [67,68]. Three yearly surveillance visits should be offered to patients diagnosed with extensive CAG or gastric IM defined as that involving the gastric antrum and body [56,69]. Additional risk factors that might warrant surveillance include a strong family history of gastric cancer or resistant H pylori infection, and more intensive follow-ups every 1–2 years might be considered. Biopsies should be taken of sites according to the Sydney protocol (Figure 4), and within these, the BSG guidelines recommend direct sampling of areas of gastric IM.

## 8. Novel Technologies/AI

The last two decades have seen a rapid expansion in the field of artificial intelligence (AI), with great advances made in applications for endoscopy. Improving the detection of neoplasia in the upper GI tract is an area well suited for automated adjuncts to both complement optical diagnosis (computer-aided diagnosis, CADx) and reduce missed lesions through improved lesion detection (computer-aided detection, CADe). The ESGE has published a 2022 position statement on the expected value of AI in gastrointestinal endoscopy [70].

### 8.1. Oesophageal Neoplasia

Several research groups have developed AI systems with high sensitivities for detecting BE-related neoplasia during real-time endoscopy (ranging from 83.7–95.4%) [71,72,73]. Furthermore, two systematic reviews and meta-analyses pooling real-time and standalone performance have also shown detection performances of 88–96% [70]. In addition to the detection of dysplasia, AI tools have been developed to discriminate submucosal invasion depth in BE-related cancer (T1a vs. T1b), with one study showing sensitivity (77%), specificity (64%) and accuracy (71%) that were not significantly different from those of experts [54]. Further research is needed to establish AI’s use in predicting submucosal invasion, but this has potentially large cost- and treatment-saving benefits.

In the setting of early SCC, a UK study trained a convolutional neural network using magnified NBI images to detect and stage early SCC based on the IPCL classification system, with 93.7% accuracy and 86.2% sensitivity and 98.3% specificity for detecting early SCC lesions [74]. Subsequently, a multicentre study developed an AI system to predict IPCL subtypes for superficial early SCC and found that the ability of junior endoscopists to diagnose IPCL subtypes was significantly improved with AI assistance [75].

### 8.2. Gastric Neoplasia

Detection of EGC presents a potentially greater challenge than in the oesophagus. Lesions are often flat and subtle, occurrence in lower-incidence countries is infrequent, and the anatomy of the stomach, with its rugae, large surface area and overlying mucus, all increase the difficulty of training an AI algorithm to detect EGC. A 2021 Chinese RCT developed an updated version of ENDOANGEL for monitoring blind spots during OGD to improve the quality of endoscopy. In a multicentre RCT, over 1000 participants undergoing upper GI endoscopy were randomly assigned to either ENDOANGEL-assisted endoscopy or routine endoscopy. The AI-assisted group had fewer blind spots and longer inspection times, and the ENDOANGEL correctly predicted EGC in all three cases [76]. Further to this, a single-centre randomised controlled trial tested an AI system designed to detect gastric neoplasia. They performed same-day tandem OGD, with patients first undergoing either AI-assisted (AI-first) or routine (routine-first) WLE. Targeted biopsies were taken of all detected lesions, with the gastric neoplasm miss rate significantly lower, fewer biopsies needed and higher positive predictive values in the AI-first group [77]. A meta-analysis of AI-assisted endoscopy for the diagnosis of EGC (16 studies) found that AI was more accurate than experts, achieving an AUC of 0.96 (95% CI, 0.94–0.97), sensitivity of 86% (95% CI, 77–92%) and specificity of 93% (95% CI, 89–96%) [78]. These are very promising data that may lead to changes in practice in the near future.

## 9. Conclusions

There is now a wealth of data to support best practices in the detection and surveillance of pre-cancerous and early cancerous lesions in the upper GI tract; national and international guidelines should be adhered to in clinical practice and form the basis of treatment decisions. Once high-risk precursors are identified, endoscopists should be familiar with what imaging adjuncts there are to maximise neoplasia detection, including chromoendoscopy and enhanced imaging. With the rapid advancements seen in newer sampling techniques and AI-assisted endoscopy, earlier detection and real-time classification of neoplasia should continue to improve; a clear understanding of the features of early neoplasia remains essential to improving early detection. We believe future work is needed to improve quality indicators in the diagnosis and surveillance of upper GI precursor lesions, especially for gastric lesions for which practice in low-incidence countries has lagged behind that of Barrett’s surveillance. Ongoing research in AI could facilitate improved quality in endoscopic detection with upper GI endoscopy, through the automated assessment of the completeness of inspection and automated recognition of lesions (BE, CAG, gastric IM).

## Figures and Tables

**Figure 1 diagnostics-14-00301-f001:**
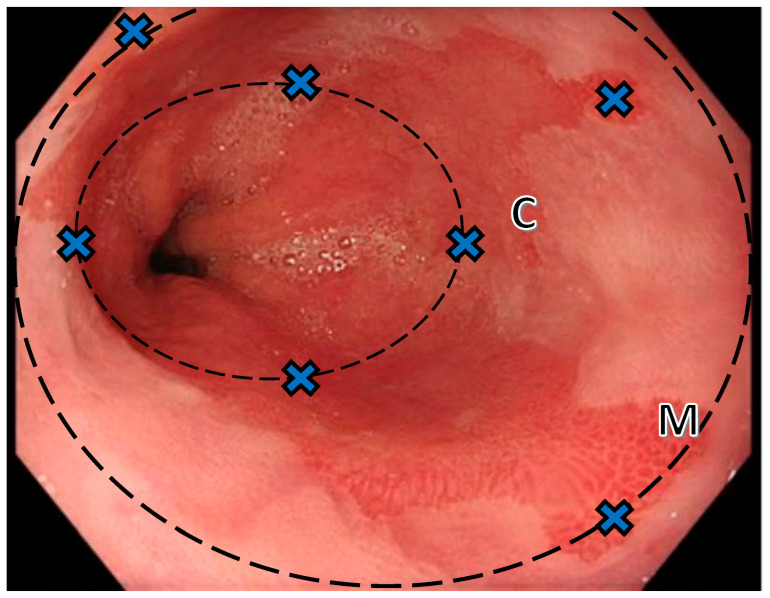
Barrett’s—Prague classification and Seattle protocol. Endoscopic appearance of Barrett’s oesophagus with biopsy locations marked with crosses to illustrate the location of biopsies. When measuring a Barrett’s segment, the Prague classification is used: first measure the length of circumferential (C) Barrett’s in cm from the GOJ (often easiest located by the top of the gastric folds), then measure the length of the maximal (M), here measured as C1M3. Seattle protocol biopsies: Four quadrantic biopsies should be taken, starting 1–2 cm above the GOJ and repeated every 2 cm. It is advisable to begin distally and work proximally to avoid views being obscure by bleeding. Where a narrow tongue is present, a single biopsy may suffice at that level; targeted biopsies of visible lesions should be taken first and sent in a separate pot.

**Figure 2 diagnostics-14-00301-f002:**
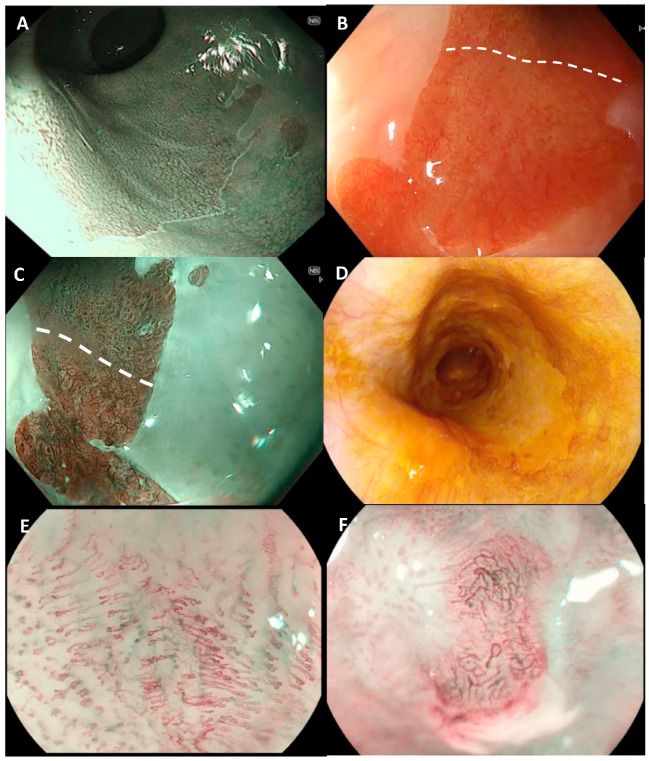
Barrett’s and early squamous neoplasia. (**A**) shows an example of a Barrett’s segment after acetic acid application. Here, the mucosal pit patterns are clearly visible, with no early loss of aceto-whitening. In (**B**), an example of flat Barrett’s neoplasia is seen in white light with near focus. Below the dashed line, the surface pattern is irregular with disorganised vessels. In (**C**), the same area is better seen with NBI and near focus; above the dashed line, the pit pattern is regular, and below there are disorganised irregular pits with irregular vessels, suggestive of high-grade dysplasia or intramucosal cancer. (**D**) shows flat squamous neoplasia after application of Lugol’s, with a long area of Lugol voiding (paler) seen at 5 o’clock. (**E**,**F**) are examples of squamous neoplasia seen with magnification; the intrapapillary capillary loops (IPCLs) can be seen. In (**E**), there are type B1 vessels with increased tortuosity and vessel density but retained loop structure corresponding to T1a carcinoma in situ, and in (**F**) there are type B2 vessels with gross dilatation and tortuosity with loss of normal loop structure, suggestive of T1a cancer with invasion of the muscularis mucosa.

**Figure 3 diagnostics-14-00301-f003:**
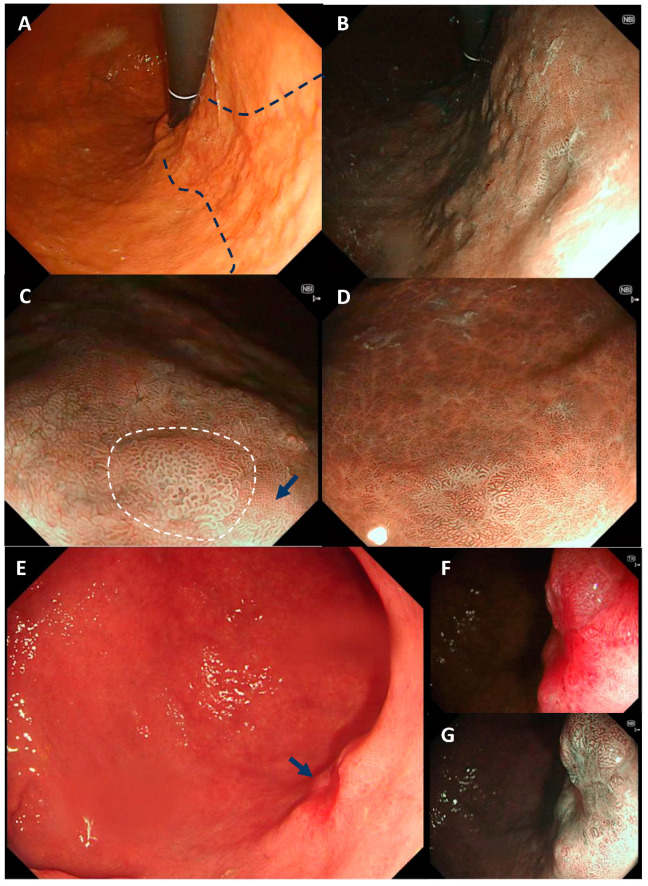
Atrophic gastritis, gastric IM and early cancer. (**A**,**B**) show the appearance in retroflexion of pan-atrophy with widespread gastric IM. With white light (**A**), there is an uneven surface with patchy erythema; in NBI (**B**), the uneven surface is accentuated and the elongated groove-type pit pattern can be seen. In (**C**,**D**), near-focus NBI allows closer interrogation of the surface pit pattern, with the white opaque substance (WOS) on IM visible in (**C**) (circled by dashed line) and adjacent normal corpus pits (blue arrow). (**D**) shows an example of multifocal gastric IM seen with NBI with near focus. In (**E**), an early gastric cancer is visible (blue arrow) along the distal greater curvature; morphologically, this is a Paris IIa and IIc lesion, with the central depression suggesting higher-grade changes. (**F**,**G**) show the same early gastric cancer with white light near-focus and NBI near-focus where the distorted vascular and surface patterns can be seen.

**Figure 4 diagnostics-14-00301-f004:**
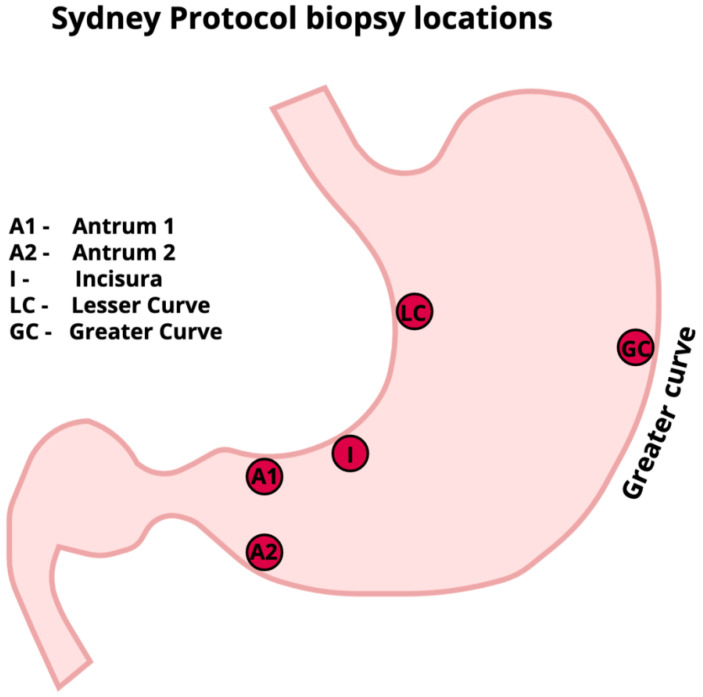
Sydney protocol for chronic atrophic gastritis. Sydney protocol biopsies should be taken in patients with endoscopic evidence of CAG and/or gastric IM; these allow histological confirmation and staging of extent to help with risk stratification and surveillance planning. Biopsies should be taken and sent in separate pots from the antrum (site A1 and A2), incisura (site I), lesser curve (site LC) and greater curve (site GC). Documentation of the location of the atrophic border if visible endoscopically, and the endoscopic staging using the modified Kimura–Takemoto system, helps standardise reporting.

## Data Availability

Data sharing not applicable.

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
