# Peer review of "Latest Advances in Endoscopic Detection of Oesophageal and Gastric Neoplasia"

_diagnostics, 2024, doi:10.3390/diagnostics14030301_

Round 1

Reviewer 1 Report

Comments and Suggestions for Authors

The authors provide a nice concise summary of the current state of endoscopic detection of esophageal/gastric neoplasias, including new AI approaches.

My only suggestion is to provide an illustration of the Prague classification/Seattle protocol for Barrett's to illustrate the scoring, as well as the other protocols mentioned for the gastric section, such as the Sydney protocol.  These visuals would be very helpful to readers/endoscopists to reference as they are reading this work.

Author Response

Many thanks for making this point about illustrating classification and biopsy protocols, we have made two new figures which are now being included in the revise manuscript, one illustrating the Prague classification and Seattle protocol and one illustrating the Sydney protocol.

Reviewer 2 Report

Comments and Suggestions for Authors

A job well done.

Author Response

Thank you for taking the time to review this manuscript.

Reviewer 3 Report

Comments and Suggestions for Authors

Waddingham et al. reviewed the current approach to diagnosis of these conditions and the latest advanced imaging and diagnostic techniques in this manuscript. The article includes current techniques and new developments. Thank you for giving opportunity to review this  this  study.

Author Response

(The authors gave the same response as above.)

Reviewer 4 Report

Comments and Suggestions for Authors

Dear Editor,

I have read the review article titled " Latest advances in endoscopic detection of oesophageal and gastric neoplasia" with great interest. The comprehensive analysis of various endoscopic techniques and their evolution over time provides invaluable insights into the detection and management of upper GI neoplasias. I particularly appreciate the in-depth discussion on the role of Barrett's Oesophagus and the various diagnostic modalities employed in its surveillance.

However, I wish to highlight a few aspects that could further enrich this review:

1.           Integration of Multidisciplinary Approaches: The review could benefit from a more detailed discussion on the role of a multidisciplinary approach, incorporating the perspectives of pathologists, oncologists, and surgeons alongside endoscopists. This would provide a more holistic view of patient management in the context of esophageal and gastric neoplasias.

2.           Real-World Application of Advanced Technologies: While the review excellently covers advanced technologies like AI, CLE, and VLE, it would be beneficial to discuss their real-world applications and accessibility in clinical practice, especially in settings with limited resources.

3.           Patient-Centric Perspectives: Including patient-centric outcomes, such as patient comfort, tolerance of procedures, and psychological impact of surveillance and diagnosis, could add another dimension to the review.

4.           Comparative Effectiveness: A comparative analysis of the effectiveness of different endoscopic techniques in various clinical scenarios would provide valuable guidance for practicing endoscopists.

5.           Cost-Effectiveness Analysis: An exploration of the cost-effectiveness of these advanced diagnostic techniques would be valuable, especially considering the financial implications on healthcare systems globally.

6.           Guidelines and Standardization: More emphasis on the need for standardized guidelines across different regions and the challenges in implementing these guidelines would be insightful.

7.           Future Directions and Research Gaps: Finally, a section dedicated to potential future directions and identifying gaps in current research could guide upcoming studies in this field.

I commend the authors for their extensive work and believe these additional aspects could make the review even more comprehensive and beneficial for its readers.

Author Response

We thank the reviewer for their detailed comments, we will address each point in turn.

  1. We have now included a section specifically addressing the importance of an MDT approach, we agree that this is a vital part of managing these conditions. This includes pathologists, surgeons and radiologists.
  2. In terms of real-world application of advanced imaging and AI it is very true that many novel or recent technologies are not widely available, in most centres there will be a single modality in use. AI is also not yet in routine clinical practice for the detection of upper GI neoplasia and remains predominantly a research tool.  Part of the role of this review is to draw attention to these technologies so the reader has a greater awareness of what is available or may be available in the near future.  We have amended the wording of the manuscript to try to reflect our practice and what we have available.
  3. A diagnosis of Barrett’s is associated with psychological morbidity, and endoscopic surveillance is part of this burden. We have included a paragraph highlighting this issue and cited some relevant data.  We feel an in-depth review of this is outside the scope of the paper.
  4. Comparative effectiveness – we have already tried to address this where data exists. There are not many studies specifically comparing imaging modalities with each other.
  5. Cost-effectiveness analysis – this has been studied in depth by NICE (National Institute of Clinical Excellence) for Barrett’s oesophagus, for example here: https://www.nice.org.uk/guidance/ng231/documents/evidence-review-5 for non dysplastic BE. We feel that an in-depth review of this is more in the realms of a meta-analysis and beyond what can be reasonably included given that this is a review focusing mainly on endoscopic detection.
  6. This is an important point and to some extent variation in guidelines reflects geographical variation in the population requiring health care e.g. disease incidence, but also the health care culture and resources that are available.
  7. Future directions - we have briefly addressed our view on this in the concluding paragraph.

We hope these amendments are satisfactory.

Round 2

Reviewer 4 Report

Comments and Suggestions for Authors

Dear editor, my suggestions were fully addressed in the revised manuscript. 

No further comments. The paper is suitable for publication.